# Search for Synergistic Drug Combinations to Treat Chronic Lymphocytic Leukemia

**DOI:** 10.3390/cells11223671

**Published:** 2022-11-18

**Authors:** Eleonora Ela Hezkiy, Santosh Kumar, Valid Gahramanov, Julia Yaglom, Arkadi Hesin, Suchita Suryakant Jadhav, Ekaterina Guzev, Shivani Patel, Elena Avinery, Michael A. Firer, Michael Y. Sherman

**Affiliations:** 1Department of Molecular Biology, Ariel University, Ariel 40700, Israel; 2Department of Chemical Engineering, Ariel University, Ariel 40700, Israel; 3Department of Mathematics, Ariel University, Ariel 40700, Israel; 4Adelson School of Medicine, Ariel University, Ariel 40700, Israel; 5Ariel Center for Applied Cancer Research, Ariel University, Ariel 40700, Israel

**Keywords:** CLL, ibrutinib, bortezomib, rapamycin, drug combination, synergy, xenograft

## Abstract

Finding synergistic drug combinations is an important area of cancer research. Here, we sought to rationally design synergistic drug combinations with an inhibitor of BTK kinase, ibrutinib, which is used for the treatment of several types of leukemia. We (a) used a pooled shRNA screen to identify genes that protect cells from the drug, (b) identified protective pathways via bioinformatics analysis of these gene sets, and (c) identified drugs that inhibit these pathways. Based on this analysis, we established that inhibitors of proteasome and mTORC1 could synergize with ibrutinib both in vitro and in vivo. We suggest that FDA-approved inhibitors of these pathways could be effectively combined with ibrutinib for the treatment of chronic lymphocytic leukemia (CLL).

## 1. Introduction

Chronic Lymphocytic Leukemia (CLL) is a B cell malignancy characterized by the progressive accumulation of monoclonal CD5/CD19 positive B lymphocytes in the peripheral blood, bone marrow, lymph nodes, and spleen [1,2]. It is the most common leukemia in adults in Western countries, with a median age of diagnosis of 70 years, and while recent developments in chemo- and immunotherapy have extended the overall survival rate of patients with CLL, the disease remains incurable [3]. Until recently, the standard treatment for CLL was a combination of chemotherapy (fludarabine/cyclophosphamide), targeted monoclonal antibody therapy (rituximab), and, for some patients, allogeneic stem cell transplantation [4]. However, many older adult (>75 years) patients and those with comorbidities do not tolerate the toxic side effects associated with these treatments and require other treatment options. In 2014, the US Food and Drug Administration (FDA) approved ibrutinib for CLL patients who have received at least one previous therapy [5]. In 2016, the approval was extended to first-line treatment of patients with CLL [6,7]. Ibrutinib is a small molecule that binds covalently to Cys-481 in the ATP binding of the kinase domain of Bruton Tyrosine Kinase (BTK), which is overexpressed and constitutively active in B cell malignancies, including CLL [8]. This binding prevents BTK from phosphorylating its substrate PLCγ, thereby halting its ability to affect downstream activation of MAPK, NFκB, and AKT pathways leading to inhibition of proliferation of the B cells [9].

In a phase I study (#NCT00849654), ibrutinib monotherapy showed a 69% overall response rate, with 16% of patients achieving a complete response [3]. Therefore currently, ibrutinib is one of the major BTK inhibitor therapies.

Despite ibrutinib’s promising activity, approximately 30% of patients have primary resistance to the drug, while other patients develop secondary resistance [3]. There are several mutations in BTK that may be responsible for ibrutinib resistance [10]. Some of the frequent mutations are at the ibrutinib binding site, leading to a reduction in BTK’s affinity to ibrutinib. Another possible reason for ibrutinib resistance is constitutively active B cell receptor (BCR) signaling, which activates signaling pathways that enhance cell survival and proliferation independently of BTK signaling.

Understanding the mechanism of ibrutinib resistance in CLL can help develop next-generation therapeutic strategies, such as drug combinations to prevent or reduce the incidence of resistance [11]. Our goal was to find complementary drugs that can enhance the anticancer activity of ibrutinib and overcome ibrutinib’s resistance to the treatment of CLL patients.

## 2. Materials and Methods

### 2.1. Cell Culture and Reagents

HL-60 (human promyeloblast, ATCC Cat# CCL-240) cells were grown in RPMI-1640 (Cat# 01-100-1A) medium supplemented with 2 mM L-Glutamine, 1 % penicillin/streptomycin (pen/strep), and 10% *v*/*v* Fetal Bovine Serum (FBS); MEC1 (human chronic lymphocytic leukemia, ACC-497, a gift from Prof. Yair Herishanu Tel Aviv University) cells were grown in Iscove’s Modified Dulbecco’s Medium (IMDM) (Cat# 01-058-1A, Sartorius, Haifa, Israel) supplemented with 2 mM L-Glutamine, 1% penicillin/streptomycin (pen/strep), and 10% *v*/*v* FBS; A20 (murine CLL) cells were grown in RPMI-1640 medium supplemented with 2 mM L-Glutamine, 0.4% pen/strep, and 10% *v*/*v* FBS. Cells were cultured in a humidified atmosphere with 5% CO^2^ at 37 °C. All cell lines were tested for mycoplasma contamination every 3 months using a commercial PCR-based kit (Sartorius, Israel). Ibrutinib was purchased from LC Laboratories (Woburn, MA, USA) (Cat# BNI-3311), MG132 from Biomol (Hamburg, Germany) (LKT-M2400.5), Bortezomib from Sigma-Aldrich (Rehovot, Israel) (Cat# 5043140001), Rapamycin (Cat# sc-491222) and Hydroxychloroquine (Cat# sc-490660) from Santa-Cruz (Enco, Petach Tikvah, Israel), Tofacitinib from TRC (Cat# C781351, Enco, Petach Tikvah, Israel), EX-527 (Cat# HY-15452), SP600125 (Cat#HY-12041), 17-AAG (Tanespimycin Cat#HY-10211), and SB29063 (Cat#HY-11068) from MCE (MedChemExpress, NJ, USA), and FX9847 was provided by Felicitex Therapeutics, Inc. (Natick, MA, USA).

### 2.2. Decipher Pooled shRNA Library

The Module 1 Decipher shRNA library (Cat. #DHPAC-M1-P) was purchased from Cellecta Inc. (Mountain View, CA, USA) and used according to the manufacturer’s recommendations, as used and reported in [12]. Briefly, the lentivirus expressing shRNA library was produced and used to infect ibrutinib-sensitive HL-60 cells, as described below.

### 2.3. shRNA Library Lentivirus Preparation

Lentivirus shRNA library was prepared using the corresponding shRNA plasmids (30 µg) (along with virus coat packaging plasmids using Lipofectamine 3000 infection reagent (Cat# L3000015, Thermo Scientific) in HEK293T cells. Briefly, cells were passaged and grown at 80–90% confluency in DMEM high glucose media supplemented with 4 mM glutamine, 1 mM sodium pyruvate, and 5% heat-inactivated FBS. For transfection, reagents were mixed in Opti-MEM (Cat# 31985070, Thermo Scientific, Waltham, MA, USA) supplemented with 4 mM glutamine and co-incubated overnight. The next day, the media was changed with Opti-MEM supplemented with 5% FBS and 4 mM glutamine and kept for a further 24 h. Viruses were harvested using a 0.45 μm filter and kept at −80 °C for further use [12].

### 2.4. Infection of HL-60 Cells with shRNA Lentivirus

HL-60 cells at a seeding density of 1.5 × 10^6^ cells/mL were infected with 20% shRNA virus titer and 1% polybrene (Cat#TR-1003, Sigma-Aldrich). We titrated the virus to obtain 15–30% infectivity, as judged by observation of cells under Olympus Fluorescent Microscope IX81 for detecting RFP-positively infected cells using Olympus cellSens Dimension software (Ver.1.18). Then, the cells were centrifuged at 250× *g* for 2.5 min at room temperature, the supernatant was discarded, and the pellet was resuspended with fresh RPMI medium. We tested the percentage of RFP signal 48 h after infection using a fluorescent microscope, as described before and propagated them. Four days after infection, the cells were divided into two groups and cultured for 24 h. The control group was cultured in a fresh medium alone, while the experimental group was cultured in a medium containing 30 µM ibrutinib. At the end of the culture period, viable cells were collected from both groups (refer to the general protocol of infection [12]).

### 2.5. Collection of Viable HL-60 Cells

Viable HL-60 cells were collected with a Percoll gradient method [13,14]. The gradient was obtained by preparing 20%, 30%, 40%, and 50% Percoll solutions in a mixture containing Dulbecco’s Phosphate Buffered Saline (DPBS) (Cat# 02-020-1A, BI Israel), 1 M HEPES buffer (Cat# 03-025-1B, BI Israel), and Trypan blue 0.4% solution (Cat#T8154, Sigma-Aldrich). About 10^7^ HL-60 cells in 2 mL DPBS were added to 2 ml of each Percoll gradient and centrifuged at 500× *g* for 15 min at room temperature. After centrifugation, a band of live cells appeared on top of the 40% phase gradient. These cells were collected and cryopreserved at −80 °C for further analysis.

### 2.6. Genomic DNA Extraction

For extracting the DNA from the cells, we used DNeasy Blood & Tissue Kit (Cat# 69504, Qiagen), according to the manufacturer’s protocols as used and reported in [12].

### 2.7. Amplification of the Barcodes

We performed the first PCR using a wide, separated primer set to a region that covers the entire region of barcodes. A second PCR reaction was then performed with primers that were closer to each other but still covered the barcode region (Appendix A, ordered from Integrated DNA Technologies (IDT), Leuven, Belgium).

The first PCR (PCR 1) was performed using a Titanium Taq DNA Polymerase (Cat# 639209, Takara Bio). Separation of the PCR products from primers and gel purification was conducted with QIAquick PCR & Gel Cleanup Kit (Cat# 28506, Qiagen, Hilden, Germany). The second PCR (PCR 2) was carried out using nested primers, either generic or having unique sample barcodes, as two rounds of nested PCR are necessary to increase the specificity of polymerization. PCR 2 was performed using Phusion High-Fidelity PCR Master Mix (Cat#F531, Thermo Scientific). Since several samples could be sequenced in one chip, all corresponding samples were multiplexed by adding an additional sample barcode during the second round of PCR. Samples were normalized individually, then pooled together, and purification of the PCR products was completed using AmpureXP magnetic beads (Cat#A63882, Beckman Coulter, Budapest, Hungary) following manufacturer protocol [12]. Next, the barcodes were sequenced using the Ion Torrent Sequencing platform.

### 2.8. Bioinformatic Pathway Analysis

Bioinformatic pathway analysis was performed using the Broad Institute software package for gene set enrichment analysis (GSEA); the pathways were denoted as being significantly enriched if they passed the threshold of the false positive hits [15,16].

### 2.9. Synergy Tests

A test for sensitivity to drug combinations was performed with three cell lines. First, we determined the subtoxic concentrations at each line by culturing them with different drug concentrations for 24–72 h. Then, we combined the subtoxic concentration of each drug with the subtoxic concentration of ibrutinib and cultured the cells with this combination for 24 h. The cytotoxic effect of drug synergy was determined by live/dead cell counts by (i) hematocytometer (EVE™ Automated Cell Counter, NanoEntek, Seoul, Korea), (ii) metabolic activity (XTT method (Catalog # 20-300-1000, Sartorius), (iii) flow cytometry cell cycle analysis (CytoFLEX S, Beckman Coulter flow cytometer) [17], (iv) flow cytometry analysis of apoptosis using annexin V binding kit (Cat#640930, Biolegend and cat#ab14085, Abcam), and (v) measuring Caspase substrate cleavage of poly(ADP-ribose) polymerase (PARP) by immunoblotting [18,19].

### 2.10. Lentivirus Preparation

We used a plasmid mCherry-2A-luciferase_ LNT-Sffv-MCS-ccdB for preparing Luc-mCherry lentivirus. Lentivirus was prepared using the Luc-mCherry plasmid along with virus coat packaging plasmids using Lipofectamine 3000 infection reagent (Cat#L3000015, Thermo Scientific) in HEK293T cells. Briefly, cells were passaged and grown at 80–90% confluency in DMEM high-glucose media supplemented with 4 mM glutamine, 1 mM sodium pyruvate, and 5% heat-inactivated FBS. For transfection, reagents were mixed in Opti-MEM (Cat#31985070, Thermo Scientific) supplemented with 4 mM glutamine and coincubated overnight. The next day, the media was changed with Opti-MEM supplemented with 5% FBS and 4 mM glutamine and kept further for 24 h. Viruses were harvested using a 0.45 μm filter and kept in −80 °C for further use.

### 2.11. A20 Cell Infection with Luc-mCherry Lentivirus

A20 cells were washed with PBS and cultured in an RPMI medium in the absence of antibiotics and serum prior to infection. Infections were performed using cells at a density of 1.5 × 10^6^ cells per ml, 10% of lentivirus Luc-mCherry titer, and 1% of polybrene (Cat# TR-1003-G, Sigma-Aldrich). The suspension was centrifuged at 500× *g* for 90 min at room temperature. Then, the suspension with the cells in pellets was transferred to a 3.5 cm plate and diluted with fresh RPMI medium to the density of 0.5 × 10^6^ cells/mL. Forty-eight hours after infection, the cells were tested by flow cytometry and Olympus Fluorescent Microscope IX81 for mCherry red signal using Olympus cellSens Dimension software. A week after infection, the mCherry-positive cells were sorted and propagated. Aliquots of A20-luc-mCherry cells were frozen at −80 °C for future use.

### 2.12. Real-Time PCR Analysis

Total RNA was isolated from cells using Qiagen RNeasy Plus Mini kit (Cat. #74134; GmbH) as used and reported in [16]. The qScript cDNA Synthesis Kit (Cat#95047-100, Quanta Bio, Beverly, MA, USA) was used to convert 1 µg mRNA into cDNA. qRT–PCR was performed using the PerfeCTa SYBR Green FastMix ROX (Cat #95073-012, Quanta Bio, Beverly, MA, USA), according to the manufacturer’s protocols (18). Expression levels of β-Actin were used as an internal control. Real-time analysis was performed with an AriaMx Real-Time PCR System (Agilent AriaMx Real-time PCR System) using the primers (Appendix A) ordered from Integrated DNA Technologies (IDT), Leuven, Belgium).

### 2.13. Immunoblotting

Cells were lysed with lysis buffer (50 mM Tris-HCl (pH 7.4), 150 mM NaCl, 1% Triton X-100, 5 mM EDTA, 1 mM Na_3_VO_4_, 50 mM β-glycerophosphate, 50 mM NaF) supplemented with Protease Inhibitor Cocktail (Cat. #P8340, Sigma-Aldrich, Rehovot, Israel) and Phenylmethylsulfonyl fluoride (PMSF). Samples were adjusted to have equal concentrations of total protein and subjected to PAGE electrophoresis followed by immunoblotting with respective antibodies [16].

### 2.14. In Vivo Assessment of Drug Combination in the Animal Model

All animals were housed under pathogen-free conditions, and the animal procedures were approved by Ariel University Institutional Animal Care and Use Committee (Permit number IL-216-01-21). Mice (BALB/c, female, 6 weeks, 17–19 g weight) were purchased from Envigo, Israel, and maintained in a controlled environment (25 °C) with free access to food and water. When the mice were 9 weeks old, 0.5 × 10^6^ A20 Luc-mCherry cells in 100 ul PBS were injected intravenously into the vein tail. A20 mCherry cell growth was evaluated by analyzing the percentage of mCherry signal in blood samples taken from the tail vein on days 11 and 24. The samples were treated with Red Blood Cell lysis buffer (Cat# 4333-57, Thermo Scientific) for 10 min at room temperature. The reaction was stopped by the addition of an equal volume of PBS, followed by centrifugation at 500× *g* for 10 min at room temperature. The cell pellet was resuspended in FACS buffer, and A20 mCherry positive cells were measured by flow cytometer. Acquired data were analyzed using FlowJo software (v10.8.1). When the mCherry signal reached 7% of total white blood cells (20 days after cells injection), the mice were randomly divided into 6 groups of 4 mice each, and drug treatments began based on drug concentrations from published protocols [20,21,22,23,24]. Group 1 mice received only vehicle (Control); Group 2 mice received a daily intraperitoneal injection of 9 mg/kg of Ibrutinib; Group 3 received 0.5 mg/kg of Bortezomib intraperitoneally twice a week; Group 4 received a combination of ibrutinib (9 mg/kg) and with bortezomib (0.5 mg/kg) intraperitoneally twice a week. Blood tests were taken eight days after treatment to evaluate the mCherry signal by flow cytometry. Body weight was monitored twice weekly for the entire span of the experiment.

## 3. Results

### 3.1. Identifying Genes Involved in the Enhancement of the Response to Ibrutinib

Firstly, we sought to identify the genes, the depletion of which protects from ibrutinib using pooled shRNA screen. We reasoned that small molecules that inhibit products of these genes might enhance the ibrutinib sensitivity of cancer cells. CLL lines showed very low efficiency of lentiviral infection, which is critical for performing the screen. Therefore, we decided to use HL-60 AML cells, which demonstrated reasonably high efficiency of the infection (15%), for the screen to predict synergistic drugs based on the screen, and then validate drug effects with standard CLL lines. Briefly, HL-60 cells were infected with the shRNA library. Two days postinfection, virus-carrying cells were selected with puromycin and then split into untreated control and ibrutinib-treated groups. Upon treatment with 30 µM ibrutinib for 24 h, the death rate was about 90%. Following the treatment, viable cells were separated from dead cells using the Percoll gradient. Genomic DNA was isolated from 40 million viable cells, and shRNA barcodes were amplified by PCR from the genomic DNA mix and sequenced by Ion Torrent. The relative representation of each shRNA barcode in the control group and treated cell populations were then compared to identify species that became underrepresented in the treated cells. Upon analysis of the screening data, our criteria in selecting sensitizing hits were (1) at least three shRNA species from five that target a gene, showing similar changes in representation in treated vs. control cell populations, and (2) a ratio of representation of shRNA species in treated versus control cells of less than 0.3. These experiments allowed us to identify “sensitizing” shRNAs that represent protective genes (Appendix A). Top 25 of identified protective genes are: CASP3, EIF4E, GPR1, HDAC9, HSP90AB1, IL6, JAK2, NFKB1, NFKB2, PARP1, PARP2, PSMA1, PSMA6, PSMA7, PSMB2, PSMB3, PSMD7, PSMD8, RPS6KA1, RPTOR, STAT2, STAT3, TNF, TRAF1, TRAF4 and TNFAIP3.

### 3.2. Pathways Involved in the Ibrutinib Response

After the data collection, we ran Gene Set Enrichment Analysis (GSEA) to identify pathways that affect sensitivity to ibrutinib. Using the Molecular Signature Database [25], several potential pathways involved in the ibrutinib response were identified, including cell cycle, PI3K-AKT-mTOR, DNA repair, Interferon pathway, IL-6-JAK-STAT3 signaling, protein processing, and protein secretion (Appendix A). The screen also uncovered signaling genes that did not belong to the GSEA-identified pathways. We decided to relax our criteria and include these genes in further consideration. This gene set was analyzed using the STRING network of protein–protein interactions. Besides pathways identified by GSEA, the STRING network revealed the significance of the proteasome pathway (Appendix A).

Next, GSEA compound analysis was performed using Drug Signatures Database (DSigDB) [26] in order to identify known small molecules that can inhibit pathways identified in the shRNA screen. Among these, the focus was on FDA-approved drugs. Overall, we composed a list of nine FDA-approved drugs that target the established pathways (Appendix A) and further tested if these drugs can sensitize the cells to ibrutinib.

### 3.3. Identifying FDA-Approved Drugs That Synergize with Ibrutinib

Firstly, using the XTT assay, we determined the maximal subtoxic concentrations of the listed drugs in HL-60 cells (Appendix A). Then, combinations of these subtoxic drug concentrations were tested with 15 µM ibrutinib. The viability tests were performed with HL-60 using XTT assay and FACS analysis (Figure 1A,B and Appendix A). Among eight listed drugs, the proteasome inhibitor MG132, the mTORC1 inhibitor rapamycin, and the autophagy inhibitor Hydroxychloroquine demonstrated clear synergy with ibrutinib. Since the latter drug demonstrated toxicity in the FACS analysis, we excluded it from further considerations.

Further validation of synergistic effects of MG132 and rapamycin with ibrutinib in HL-60 cells and CLL cell lines MEC1 and A20 was performed using apoptosis Annexin IV assay (Figure 1C, Appendix A). Each of the cell lines was incubated with a subtoxic concentration of MG132 or rapamycin alone or in combination with 15–25μM ibrutinib for 24 h, and the viability of the cells was measured. A strong synergy between ibrutinib and MG132 was seen in all three cell lines (Figure 1C). The rate of apoptosis in cells incubated with ibrutinib alone was around 15–40%, depending on the cell line, while the rate of apoptosis in cells treated with the combination of ibrutinib and MG132 was around 90–94%. The synergy was confirmed on HL-60 and MEC1 cells by calculating the combination index (CI) (Appendix A) using the Loewe additivity equation and isobologram analysis [27]. There was also a synergy between ibrutinib and rapamycin. The fraction of apoptotic cells treated with rapamycin and ibrutinib together reached 54–88%.

Similar results were obtained when the cell death was measured by the PARP cleavage (Figure 1D,E). Altogether, these data indicate that certain drugs predicted on the basis of shRNA screen results can indeed synergize with ibrutinib in killing AML and CLL cell lines.

### 3.4. Synergistic Effects of the Proteasome Inhibitor Bortezomib and Ibrutinib in the Mouse CLL Model

Further, we tested a combination of ibrutinib with the FDA-approved proteasome inhibitor bortezomib in the xenograft CLL model. Briefly, BALB/c mice were IV injected with 5 × 10^5^ A20 cells expressing Luc-mCherry. When the fraction of mCherry-labeled cells in the blood reached 6–8%, we started the treatments, as described in Materials and Methods. We had four groups of mice, including (1) a nontreated control group, (2) an ibrutinib-treated group (daily 9 mg/kg), (3) a bortezomib-treated group (0.5 mg/kg twice a week), and (4) a combination group treated with ibrutinib 9 mg/kg daily and bortezomib 0.5 mg/kg twice a week. Blood samples were taken on days 11 and 24 after the beginning of treatment, and the percentage of mCherry-positive cells in the blood was measured using flow cytometry (Figure 2A).

In the control group (blue curve), A20-mCherry cells continuously propagated in mice, and their presence in the blood over the course of the experiment increased almost fourfold. In the groups of ibrutinib alone (red curve) and bortezomib alone (yellow curve) treated mice, we observed an initial increase in the number of mCherry A20 cells by day 11, which was followed by a decrease, so that at the end of the experiment the number of A20 cells was similar to that at the starting day of treatment. However, in the combination group (ibrutinib + bortezomib) (green curve), the treatment led to a steady, almost fourfold decrease in the mCherry-labeled A20 cells (Figure 2B).

Therefore, both in vitro and in vivo experiments demonstrate the clear benefits of combining ibrutinib with a proteasome inhibitor in the CLL model.

### 3.5. Possible Synergy Pathways

We further attempted to address the mechanisms of synergy between ibrutinib/bortezomib and ibrutinib/rapamycin. The complication was that each of the treatments could affect multiple signaling pathways, both pro- and antiapoptotic; therefore, multiple effects might be involved. One can envision a scenario that two drugs can enhance the activity of a proapoptotic pathway in a synergistic way or reduce the activity of an antiapoptotic pathway in a synergistic way. Alternatively, the drugs can affect distinct pathways (either pro- or antiapoptotic), and their altered balance at a very downstream stage facilitates the execution of apoptosis.

We considered both possibilities in the search for potential players in apoptosis-regulating pathways that may play a role in the synergy. To search for synergy within an apoptosis-related signaling pathway, we extracted published RNAseq data from the GEO database on the gene expression effects of ibrutinib in the HL-60 cell line and performed GSEA analysis to identify pathways activated and downregulated by the drug (Appendix A). Among ibrutinib-activated pathways, several pathways were found that may be relevant to either activation or suppression of apoptosis, including TNF/inflammation, Myc, STAT3, and unfolded protein response (UPR). Therefore, we tested the potential synergistic effects of the combination of ibrutinib and MG132 on the activity of these pathways using RT-PCR of the corresponding target genes (Appendix A). We observed a clear synergy upon evaluation of the components of the UPR pathway. In the UPR pathway, three downstream targets were tested, including transcription factors ATF3, ATF4, and CHOP in HL-60 and MEC1 (Figure 3). In line with the RNAseq data, ibrutinib demonstrated moderate enhancement of expression of these targets. Surprisingly, and contrary to published results with other cells, MG132 alone did not activate the pathway in these lines [28]. However, together the two drugs showed strong synergistic activation of the CHOP expression and a somewhat weaker, but still significant, synergistic activation of the ATF3 expression in HL-60 and MEC1 cell lines (Figure 3A, B). ATF4 did not demonstrate these effects.

Since CHOP is a major proapoptotic factor, these synergistic effects on the pathway may contribute to the overall synergistic effects of ibrutinib and proteasome inhibition.

In another example of a pathway potentially involved in the synergy between ibrutinib and MG132, we probed the effects of these drugs on Akt, a component of a well-known antiapoptotic pathway [29]. We found that MG132 activates this pathway (Figure 3C), which might partially mitigate proapoptotic effects of the proteasome inhibition, e.g., via activation of JNK, stabilization of p53, or another proapoptotic effect. The addition of ibrutinib suppressed the Akt pathway, and the addition of MG132 in the background of ibrutinib did not reverse the effects of ibrutinib on the Akt activity (Figure 3). Accordingly, synergy in the activation of apoptosis between ibrutinib and MG132 could involve inhibition of Akt by ibrutinib, which shifts the balance of pro- and antiapoptotic effects of MG132 towards enhanced apoptosis. Obviously, effects on other pathways may also contribute to the synergy.

## 4. Discussion

Previously, there have been studies to identify drugs that can synergize with ibrutinib and/or overcome ibrutinib resistance [3]. Most of them were based on prior knowledge of the mechanisms of action of potential complementary drugs. Here, we approached this problem in an unbiased way by using pooled shRNA screen to identify genes that affect sensitivity to ibrutinib. On the basis of the screen, several pathways were identified, the inhibition of which was predicted to enhance the cell’s sensitivity to ibrutinib. Then, we validated these hypotheses by testing the effects of combinations of ibrutinib with inhibitors of these pathways on the viability of AML and CLL cell lines. A synergistic combination of ibrutinib and a proteasome inhibitor was successfully validated in a mouse model of CLL.

Using a combination of a proteasome inhibitor with ibrutinib has already been suggested for multiple myeloma, and currently, there is an ongoing corresponding clinical trial [30]. Using this combination was originally suggested based on the simple fact that both drugs are used for the treatment of multiple myeloma. Our finding with the synergy between these drugs provides an experimental basis for the trial. Furthermore, it suggests that there is a rationale for testing this combination on CLL and AML patients.

Previously, it has been shown that CC-115, an inhibitor of mTOR and DNA-PK, chemically dissimilar from rapamycin, can effectively kill CLL cells resistant to ibrutinib both in preclinical models and in the clinical setting [31]. It is unclear, however, if the anti-CLL activity of CC-115 is related to the inhibition of mTOR or DNA-PK. In any case, it was not used in combination with ibrutinib. Here, we demonstrate that the combination of ibrutinib with rapamycin synergistically kills CLL cells and thus may be effective in cancer treatment. Interestingly, at least MEC1 cells are highly resistant to ibrutinib, and thus the activity of rapamycin on these cells could be considered as overcoming the ibrutinib resistance. However, rapamycin alone showed a minimal effect, and only the combination with ibrutinib demonstrated effective cell killing. Accordingly, this is a true synergy between the drugs. Overall, this study identified synergistic drug combinations with ibrutinib in an unbiased way. More extensive preclinical and clinical studies may pave the way for these combinations in the clinic.

The mechanisms of synergy between proteasome inhibitors, rapamycin, and ibrutinib have not been clarified because of the complexity of the system. Indeed, these drugs affect multiple pro- and antiapoptotic pathways, and the outcome of treatments depends on their combinations. So far, we uncovered a proapoptotic pathway that is synergistically activated by ibrutinib and MG132 and an antiapoptotic pathway inhibited by rapamycin, which might enhance cell death in response to ibrutinib. Overall, a balance of multiple pathways might define the fate of cancer cells upon the execution of apoptosis.

Our data support the idea that CRISPR or shRNA screening is a powerful tool for identifying synergistic drug combinations. The main efforts to design synergistic drug combinations so far have been modeled on the basis of multiple OMICS approaches, including –protein interaction networks, cMAP, etc. We think that incorporating data from CRISP or shRNA screens in such modeling would introduce a functional component and thus significantly enhance the approach.

## 5. Conclusions

To conclude, we integrated the strategy of a powerful genetic screen to identify the potentially druggable pathways in this study. Further, this information was utilized to rationally design the synergistic drug combinations in the animal model.

## Figures and Tables

**Figure 1 cells-11-03671-f001:**
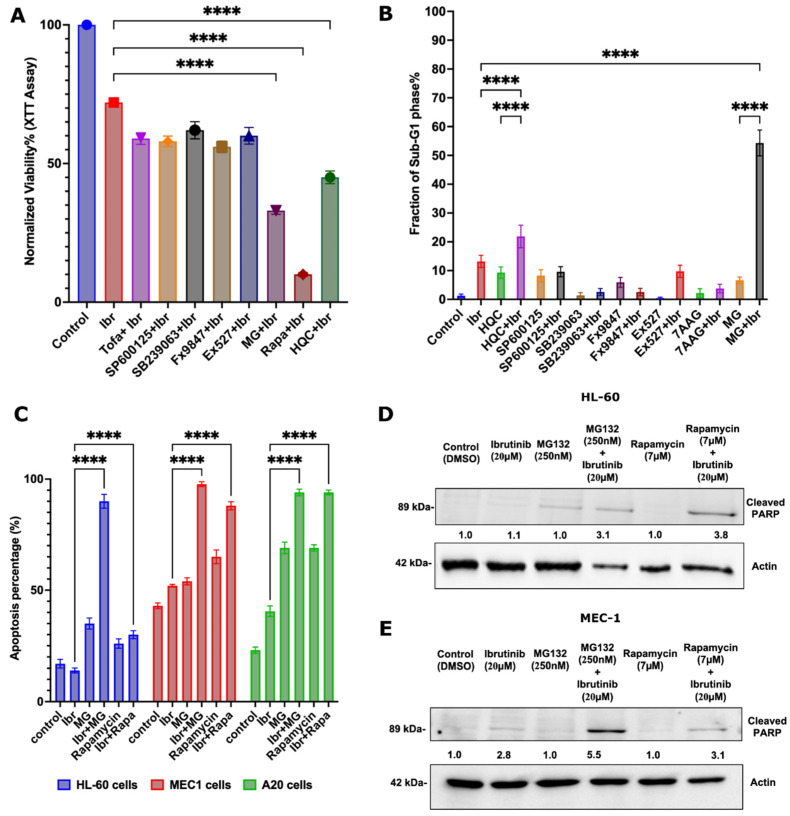
Synergy tests in vitro with HL-60, MEC1, and A20 cells. (**A**). Effects of drugs on ibrutinib sensitivity of HL-60 cells tested by XTT. Cells were incubated with 15 µM ibrutinib alone or combined with subtoxic concentrations of listed inhibitors for 24 h. The experiment was performed in biological replicates (*n* = 4), (**B**). Effects of drugs on ibrutinib sensitivity of HL-60 cells tested by FACS analysis (sub-G1 population). The experiment was performed in biological triplicates (*n* = 3), (**C**). Effects of the combination of ibrutinib with MG132 and ibrutinib with rapamycin on cell viability (HL-60, MEC-1, and A20) measured by annexin V apoptosis assay performed in biological triplicates (*n* = 3). (**D**). Effects of the combination of ibrutinib with MG132 and Ibrutinib with rapamycin tested by PARP cleavage. HL-60 and (**E**) MEC1 cells were incubated with drugs for 24 h. The experiment was performed in biological triplicates (*n* = 3) for all the above experiments. PARP cleavage was assessed by immunoblotting. Statistical analysis was performed using two-way ANOVA. All statistical analysis was performed using Graphpad (v9), level of significance was taken as (“ns” = 0.1391, **** “*p* < 0.0001”) as denoted appropriately on the above graphs.

**Figure 2 cells-11-03671-f002:**
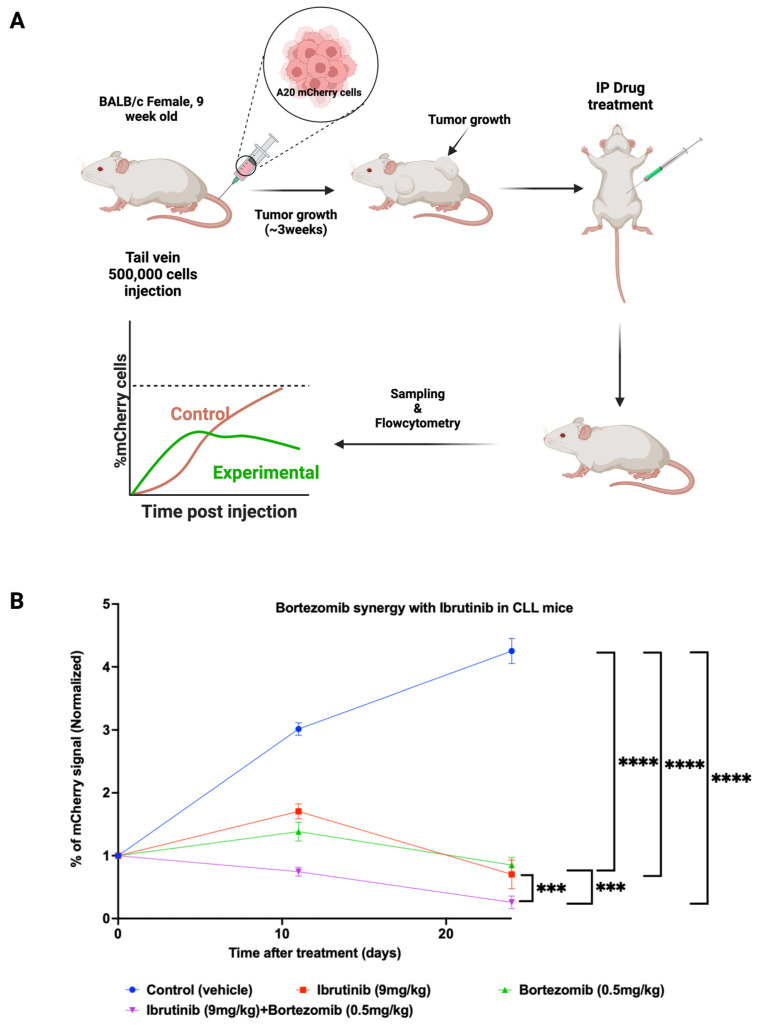
In vivo effects of ibrutinib and bortezomib on A20 leukemia. (**A**). Representative scheme of the animal experiment where 0.5 × 10^6^ A20 labeled Luc-mCherry cells were implanted by the tail vein injection. Three weeks later, the mice were treated with (1) vesicle, (2) Ibrutinib 9 mg/kg, (3) Bortezomib 0.5 mg/kg twice a week, and (4) combined Ibrutinib with Bortezomib. The influence of the drug treatments was tested by sampling the blood taken from the tail vein at two time points and analyzing it by FACS (**B**). Percentage of mCherry-positive A20 cells in the blood of mice treated for 11 and 24 days; blue curve group treated with vehicle, red curve treated with ibrutinib, yellow curve treated with bortezomib, and green curve treated with ibrutinib and bortezomib together. Each group had *n* = 4 mice; the error bar represents the standard deviation between the replicates in each group. Statistical analysis was performed using two-way ANOVA. All statistical analyses were performed using Graphpad (v9), level of significance was taken as (*** “*p*” < 0.0002, **** “*p”* < 0.0001) as denoted appropriately on the above graphs.

**Figure 3 cells-11-03671-f003:**
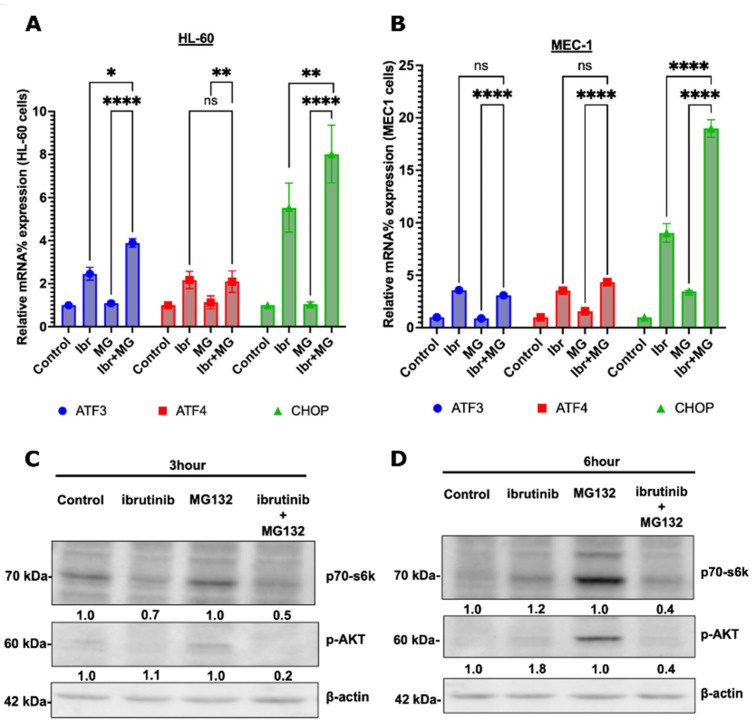
Effects of Ibrutinib and MG132 on UPR and Akt pathways. (**A**,**B**) Expression of CHOP and ATFs mRNA in response to ibrutinib and MG132 in (**A**) HL-60 and (**B**) MEC1. The cells were incubated for 12 h with ibrutinib or MG132 or their combination, and expression of the indicated species of mRNA was measured by qPCR. The experiment was performed in biological triplicates (*n* = 3). Statistical analysis was performed using two-way ANOVA. All statistical analyses were performed using Graphpad (v9), and the level of significance was taken as (“ns < 0.1234”, * “*p*” < 0.0332, ** “*p*” < 0.0021”, **** “*p*” < 0.0001), as denoted appropriately on above graphs. (**C**) Effects of MG132 and ibrutinib on phosphorylation of Akt and the mTORC1 substrate p70 S6 kinase (Thr421/Ser424). The cells were incubated with 15 µM ibrutinib and/or 100nM MG132 for 3 h and (**D**) 6 h. The phosphorylation state of the proteins was assessed by immunoblotting.

## Data Availability

All the data presented in the manuscript is available either in the text or in the Appendix A.

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
