# Peer review of "Search for Synergistic Drug Combinations to Treat Chronic Lymphocytic Leukemia"

_cells, 2022, doi:10.3390/cells11223671_

Round 1

Reviewer 1 Report

The aim is original and study well done.

for the introduction no reference to ibrutinib is made it's missing

two problems:

1) it's done on cell lines and animal model not in humans so the title can be missleading the readers

2) the toxicity is mentioned in the introduction as a main problem for treatment but is not tackle by the sudy.

Author Response

Reviewer 1:

1) it's done on cell lines and animal model not in humans so the title can be misleading the readers

The title is changed

2) The toxicity is mentioned in the introduction as a main problem for treatment but is not tackle by the study.

Toxicity is just mentioned in the Introduction. In this work, we have not reached the stage of human testing. We focused on finding complementary drugs and showed that they enhance cancer cells’ killing both in vitro and in vivo.

Reviewer 2 Report

This manuscript entitled “Using genetic screen to develop drug combinations for chronic lymphocytic leukemia” explored synergistic drug combinations for the treatment of Chronic Lymphocytic Leukemia. Eleonora et al used a pooled shRNA screen and bioinformatics analysis to find genes and pathways that related to the drugs. Then, they found inhibitors of proteasome and mTORC1 could synergize with ibrutinib. Generally, the studies in this manuscript have some implications. However, there are some serious problems with this manuscript as below
Major
1. Some figures have low resolutions, and the figures’ fonts and sizes were not uniform and clear, such as Figure 1D and E, the actin is very faint and unclear and β-actin is not consistent for every band; in figure 1 legend, there was no any A20 cell line data but the title showed it; in the figure 3 C and D, the font size was not uniform; “p“ should be “p“; also, the axis name size was too small; According to the results, the authors cannot draw the right conclusion thought the data of western blots. The authors should repeat all western blots.
2. Remarkably, although the authors concluded that there was a synergy effect of ibrutinib and proteasome inhibitor/rapamycin, it might be an additive effect. It’s more convincing to calculate a combination index (<1) using the software. In addition, the KO combination of these genes and their downstream components in vivo may provide evidence for their results.
3. In Result 1, identifying genes involved in the enhancement of the response to ibrutinib. In this part, the authors did not show which genes are involved in the enhancement of the response to ibrutinib. Moreover, they should present data in text, not only in the supplementary part.  
4. The authors could supplement cellular experiments to support whether the combined effect of ibrutinib and MG132 after the knockdown of any ATF3, ATF4, or CHOP in CLL cells, would be more convincing and interesting.
5. Some key hub genes/pathways such as UPR, ATF3, ATF4, and CHOP should be explained in the first appearance and in the discussion.
6. Again, the discussion should address the manuscript’s novelty and limitation; is there any limitation of your method and experiment design? How the combination could improve the standard strategy in the future?
Minor
1. The title of this paper must be changed. It is imprecise. Though the authors screened genes that related to drugs. Through reading the manuscript, I do not think that the authors get a clear conclusion that which genes were related to drugs.
2. The discussion is too short and general to the previous study like drug resistance, and listed the results again, explained very simply. I recommended the author could include more information as the why there was a synergy effect of proteasome inhibitor/mTOR inhibitor. How did they alter the apoptotic pathway?

Author Response

  1. We corrected these drawbacks.
  2. As requested, we provide combination index calculation (Table S7)

  3. We added the list of the top 25 genes in the text.
  4. We did this experiment, and the results were inconclusive since we could not achieve more than 50% of the gene depletion. On the other hand, we specifically note in Results and Discussion, that there are probably multiple effects leading to the synergy since multiple signaling pathway are affected. Accordingly, depletion of just one pathway may not be enough or could be compensated by changes in other pathways.

  5. As suggested, we explain the UPR abbreviation. We also note that ATF3, ATF4 and CHOP are UPR-regulated transcription factors.

  6. As suggested, we added this discussion

Minor

  1. The title was changed
  2. Added to the discussion

Reviewer 3 Report

Interesting article with average novelty. Suggestions:

1-Rationale to use rapamycin and bortezomid in combination with ibrutinib? How does the combination of drugs affected bone marrow cellularity and complete blood count?
2- How this study differ from "Rapamycin shows anticancer activity in primary chronic lymphocytic leukemia cells in vitro, as single agent and in drug combination"
Aleskog et al Leuk Lymphoma 2008 Dec;49(12):2333-43.
3-Please clarify instances where drug combination were superior than single agent in the treatment of CLL.
4- Does Bortezomib and rapamycin, without ibrutinib work in the mice model?
5- How does rapamycin affect CLL cell growth when these are resistant to ibrutinib?

Author Response

  1. The rationale to combine ibrutinib with bortezomib or rapamycin was based on the results of pooled shRNA screen that showed that depletion of genes belonging to these pathways enhances ibrutinib-induced HL-60 cell killing.

    We did not test complete blood count and bone marrow cellularity, but the overall toxicity in the mouse model was tolerable, while the count of cancer cell went down significantly.

  2. In the published study, rapamycin was tested with various drugs but not with ibrutinib. Because of the common use of ibrutinib for treatment of CLL finding that it synergizes with rapamycin is an important advance.

  3.  

    Drug combinations rather than single agents are routinely used for cancer treatment. For example: Venetoclax in combination with obinutuzumab increased overall response of CLL patients (1). Chlorambucil in combination with prednisone increased survival rate in the patients more than mono-therapy with chlorambucil (2). Also combination of cyclophosphamide, vincristine sulfate and prednisone, used to treat CLL (3).

    Reference

    1. Fischer K, Al-Sawaf O, Fink AM, Dixon M, Bahlo J, Warburton S, Kipps TJ, Weinkove R, Robinson S, Seiler T, Opat S, Owen C, López J, Humphrey K, Humerickhouse R, Tausch E, Frenzel L, Eichhorst B, Wendtner CM, Stilgenbauer S, Langerak AW, van Dongen JJM, Böttcher S, Ritgen M, Goede V, Mobasher M, Hallek M. Venetoclax and obinutuzumab in chronic lymphocytic leukemia. Blood. 2017 May 11;129(19):2702-2705. doi: 10.1182/blood-2017-01-761973. Epub 2017 Mar 21. Erratum in: Blood. 2017 Jul 13;130(2):232.
    2. Payandeh M, Sadeghi M, Sadeghi E. Cholorambucil versus Cholorambucil Plus Prednisolone as First-Line Therapy of Chronic Lymphocytic Leukemia in West of Iran. Iran J Cancer Prev. 2015 Mar-Apr;8(2):94-9.
    3. Kem R, Solimando DA, Waddell JA. CVP (Cyclophosphamide, Vincristine, and Prednisone) Regimen for Chronic Lymphocytic Leukemia and Small Lymphocytic Lymphomas. Hospital Pharmacy. 2003;38(12):1126-1133. doi:10.1177/001857870303801203

     4.

    After 24 days of bortezomib treatment we can see a decrease in the amount of tumor cells but the decrease was less significant than in ibrutinib + bortezomib group.

     5. 

    At concentrations below 5µM, rapamycin alone showed very low toxicity in MEC1, A20 (CLL) and HL-60 cells (AML). However, in combination with ibrutinib, rapamycin causes very significant cell killing (Fig. 3S).

Round 2

Reviewer 1 Report

the title as been changed as expected

the introduction and discussion were changed in part as required.

For me the paper is now good for publication

Reviewer 2 Report

Authors have better addressed all my concerns